# Expert Consensus: Main Risk Factors for Poor Prognosis in COVID-19 and the Implications for Targeted Measures against SARS-CoV-2

**DOI:** 10.3390/v15071449

**Published:** 2023-06-27

**Authors:** Francisco Javier Candel, Pablo Barreiro, Miguel Salavert, Alfonso Cabello, Mario Fernández-Ruiz, Pedro Pérez-Segura, Jesús San Román, Juan Berenguer, Raúl Córdoba, Rafael Delgado, Pedro Pablo España, Ignacio Alberto Gómez-Centurión, Juan María González del Castillo, Sarah Béatrice Heili, Francisco Javier Martínez-Peromingo, Rosario Menéndez, Santiago Moreno, José Luís Pablos, Juan Pasquau, José Luis Piñana

**Affiliations:** 1Clinical Microbiology & Infectious Diseases, Transplant Coordination, Hospital Clínico Universitario San Carlos, 28040 Madrid, Spain; franciscojavier.candel@salud.madrid.org; 2Regional Public Health Laboratory, Infectious Diseases, Internal Medicine, Hospital General Universitario La Paz, 28055 Madrid, Spain; 3Department of Medical Specialities and Public Health, Universidad Rey Juan Carlos, 28922 Madrid, Spain; jesus.sanroman@urjc.es (J.S.R.); fmperomingo@salud.madrid.org (F.J.M.-P.); 4Infectious Diseases, Internal Medicine, Hospital Universitario y Politécnico La Fe, 46026 Valencia, Spain; salavert_mig@gva.es; 5Internal Medicine, Hospital Universitario Fundación Jiménez Díaz, 28040 Madrid, Spain; acabello@fjd.es; 6Unit of Infectious Diseases, Hospital Universitario “12 de Octubre”, Instituto de Investigación Sanitaria Hospital “12 de Octubre” (imas12), Centro de Investigación Biomédica en Red de Enfermedades Infecciosas (CIBERINFEC), Instituto de Salud Carlos III (ISCIII), 28041 Madrid, Spain; mario.fernandez@salud.madrid.org; 7Medical Oncology, Hospital Clínico Universitario San Carlos, 28040 Madrid, Spain; pedro.perez@salud.madrid.org; 8Instituto de Investigación Sanitaria Gregorio Marañón (IiSGM), Centro de Investigación Biomédica en Red de Enfermedades Infecciosas (CIBERINFEC), 28007 Madrid, Spain; jbb4@me.com; 9Haematology and Haemotherapy, Hospital Universitario Fundación Jiménez Díaz, 28040 Madrid, Spain; raul.cordoba@fjd.es; 10Clinical Microbiology, Hospital Universitario “12 de Octubre”, Instituto de Investigación Sanitaria Hospital “12 de Octubre” (imas12), 28041 Madrid, Spain; rafael.delgado@salud.madrid.org; 11Pneumology, Hospital Universitario de Galdakao-Usansolo, 48960 Vizcaya, Spain; pedropablo.espanayandiola@osakidetza.eus; 12Haematology and Haemotherapy, Hospital General Universitario Gregorio Marañón, 28007 Madrid, Spain; ignacioalberto.gomez@salud.madrid.org; 13Emergency Department, Hospital Clínico Universitario San Carlos, 28040 Madrid, Spain; juanmaria.gonzalezdel@salud.madrid.org; 14Intermediate Respiratory Care Unit, Hospital Universitario Fundación Jiménez Díaz, 28040 Madrid, Spain; sarah.heili@gmail.com; 15Geriatrics, Hospital Universitario Rey Juan Carlos, 28933 Madrid, Spain; 16Pneumology, Hospital Universitario y Politécnico La Fe, 46026 Valencia, Spain; rosmenend@gmail.com; 17Infectious Diseases, Hospital Universitario Ramón y Cajal, 28034 Madrid, Spain; smguillen@salud.madrid.org; 18Rheumatology, Hospital Universitario “12 de Octubre”, Instituto de Investigación Sanitaria Hospital “12 de Octubre” (imas12), 28041 Madrid, Spain; jlpablos@h12o.es; 19Infectious Diseases, Hospital Universitario Virgen de las Nieves, 18014 Granada, Spain; jpasquau@gmail.com; 20Haematology and Haemotherapy, Hospital Clínico Universitario de Valencia, 46010 Valencia, Spain; jlpinana@gmail.com

**Keywords:** COVID, SARS-CoV-2, risk factors, poor prognosis, therapy

## Abstract

The clinical evolution of patients infected with the Severe Acute Respiratory Coronavirus type 2 (SARS-CoV-2) depends on the complex interplay between viral and host factors. The evolution to less aggressive but better-transmitted viral variants, and the presence of immune memory responses in a growing number of vaccinated and/or virus-exposed individuals, has caused the pandemic to slowly wane in virulence. However, there are still patients with risk factors or comorbidities that put them at risk of poor outcomes in the event of having the coronavirus infectious disease 2019 (COVID-19). Among the different treatment options for patients with COVID-19, virus-targeted measures include antiviral drugs or monoclonal antibodies that may be provided in the early days of infection. The present expert consensus is based on a review of all the literature published between 1 July 2021 and 15 February 2022 that was carried out to establish the characteristics of patients, in terms of presence of risk factors or comorbidities, that may make them candidates for receiving any of the virus-targeted measures available in order to prevent a fatal outcome, such as severe disease or death. A total of 119 studies were included from the review of the literature and 159 were from the additional independent review carried out by the panelists a posteriori. Conditions found related to strong recommendation of the use of virus-targeted measures in the first days of COVID-19 were age above 80 years, or above 65 years with another risk factor; antineoplastic chemotherapy or active malignancy; HIV infection with CD4+ cell counts < 200/mm^3^; and treatment with anti-CD20 immunosuppressive drugs. There is also a strong recommendation against using the studied interventions in HIV-infected patients with a CD4+ nadir <200/mm^3^ or treatment with other immunosuppressants. Indications of therapies against SARS-CoV-2, regardless of vaccination status or history of infection, may still exist for some populations, even after COVID-19 has been declared to no longer be a global health emergency by the WHO.

## 1. Introduction

The Coronavirus Infectious Disease starting in 2019 (COVID-19) has already affected more than 10% of the global population and has caused around 1% of deaths among cases, according to recorded data that have probably underestimated the real figures [1,2,3].

The clinical manifestations of infection by the Severe Acute Respiratory Coronavirus type 2 (SARS-CoV-2) range from asymptomatic infection—in around 15% of infections—to multiple organ failure and death. Most symptomatic infections are limited to catarrhal symptoms, including mild pulmonary involvement, in around 80% of cases; severe pneumonia with hypoxia in around 15% of cases; and respiratory distress, shock or multiple organ failure in around 5% of cases [4,5]; overall fatality rate is today below 0.5%. The likelihood of severe disease has changed over the course of the pandemic, mainly for four reasons: the protective effect of vaccination [6], the selection for less lethal SARS-CoV-2 virus variants [7], the lower severity of reinfections and the availability of treatments to reduce the risk of progression of COVID-19 [8]. From the end of 2020, vaccination against SARS-CoV-2 has been the primary epidemiological determinant of change in hospitalization and death, and of transmission, to a lesser extent [9].According to the epidemiological studies carried out in the pre-vaccination era, the main risk factors for poor clinical outcome in patients with COVID-19 are being aged above 65 years, certain comorbidities [10,11,12,13] and different types of immunosuppression [14,15,16,17,18,19,20,21,22,23]. In vaccinated persons of all ages, a history of SARS-CoV-2 infection has been identified as one of the main protective factors against severe COVID-19 [18]. While greater body mass index (BMI) does appear to play an independent role, so do other factors associated with obesity, such as carbohydrate intolerance, diabetes, hyperlipidemia, or a sedentary lifestyle [24,25,26]. Evidence shows that a poorer prognosis may be expected in case of pulmonary [27,28], cardiovascular [29], kidney, liver and certain degenerative neurological diseases [30]. The degree of frailty and the need for care associated with these pathologies, together with age and various debilitating chronic diseases, may be the common reason for this poorer prognosis [31]. Immunosuppression is also a recognized risk factor for severe COVID-19 or death [32,33,34,35]. The analysis of the post-vaccination period continues to show that the elements that weigh most heavily in the stratification of the individual risk of a poor post-SARS-CoV-2 infection clinical outcome are vaccination status, history of infection, age, burden of comorbidities and immunosuppression [9,36,37,38,39].

It is not easy to establish the individual weight of many of the above-mentioned factors in the severity of COVID-19. Firstly, it would be necessary to ascertain whether each factor affects only the severity of the infection or whether it also contributes to a higher risk of death. Comorbidities are often analyzed in a general way, using definitions that include diseases with very different prognoses and at different stages of severity. Many factors identified as risk factors are associated with other comorbidities, producing statistical associations that do not always indicate causality. It is also necessary to analyze whether the effect of a particular disease is due to the actual disease process, to associated pathogenic factors or to the effect of certain drugs commonly used for its treatment. Chronic diseases that lead to frailty and dependency make patients more infection-prone. In addition, most analyses of prognostic factors come from retrospective studies, with the limitations that this entails in drawing conclusions [9]. Finally, despite the existence of current predictive scales with a well-adjusted number-needed-to-treat (NNT) to select patients with SARS-CoV-2 infection who would potentially benefit from a therapeutic measure in addition to the standard treatment [39,40,41,42,43], we still do not know what the weight of each one of these factors is in the current post-vaccination era.

## 2. Therapeutic Virus-Targeted Measures against SARS-CoV-2

SARS-CoV-2 disease is characterized by a defined pathochrony that conditions the clinical presentation of its different manifestations in the patient [4,44]. The early stages are dominated by viral replication, during which the patient presents general and respiratory tract symptoms like any other viral illness, such as fever, cough, dyspnea, odynophagia, asthenia and myalgia, headache, chills, vomiting and diarrhea. In some patients (15%), this initial phase leads to an inflammatory process because of an aberrant host immune response, causing diffuse alveolar damage and leading to the development of adult acute respiratory distress syndrome (ARDS). Typically, the first phase of the disease lasts for 7 days, although both phases may present simultaneously, intertwined or successively, and viral replication can be maintained in more severe and immunosuppressed patients for weeks [4,44].

From a therapeutic standpoint, it is important to discern whether the patient has viral replication for the use of additional virus-targeted measures (AVTMs), or has a state of proinflammation indicating the use of immunomodulators. The AVTMs currently available reduce the relative risk of progression by between 50% and 90% [45,46,47,48].

Neutralizing monoclonal antibody (mAb) technology, as a therapeutic alternative to previous emerging pathologies, has developed rapidly in recent years thanks to a combination of breakthroughs in single-cell genomic analysis techniques. These techniques make it possible to obtain the sequences of variable regions of B lymphocytes isolated from individuals exposed to different infectious agents, together with the availability of electron cryo-microscopy, which facilitates the precise characterization of the epitopes recognized by the antibodies [49,50,51,52,53,54].

The classification of the mAbs used against SARS-CoV-2 depends on the antibody binding site and mechanism. Depending on epitope recognition and binding mode, RBD-specific neutralizing mAbs are classified into four main classes (I, II, III and IV) [55]: (i) Class I and II neutralizing mAbs bind to the RBM region of the RBD. The mAbs that block this RBM-ACE2 interaction are “ACE2 blockers”. Class I mAbs bind to RBD only in the upward conformation, while class II mAbs block ACE2 binding and recognize both up and down RBD; (ii) Class III neutralizing mAbs block the ACE2 binding site outside the RBM and recognize both up and down RBD; and (iii) Class IV mAbs do not overlap with the ACE2 binding site and bind to a conserved region in RBD in the “up” conformation.

Therapeutic mAbs for the treatment of COVID-19 have been developed at an accelerated pace unprecedented for any other disease. Currently, the US Food and Drug Administration (FDA) and the European Medicines Agency (EMA) have a number of mAbs approved for clinical use, primarily in the early stages of infection and even as post-exposure prophylaxis, where antiviral treatment has been shown to be most effective in COVID-19. This list is changing and currently includes bamlanivimab (LY-CoV555) plus etesevimab (LY-CoV016), casirivimab (REGN10933) plus imdevimab (REGN10987), sotrovimab (VIR-7831) and bebtelovimab (LY-CoV1404) [56]. The combination of two human antibodies, tixagevimab and cilgavimab, has been cleared for use as pre-exposure prophylaxis in people at risk of severe COVID-19, in case of absence or poor response to vaccination [57]. A subsequent clinical trial [58] has demonstrated this combination’s efficacy at preventing fatal COVID-19 in high-risk patients with early symptoms; the drug may be ideal in ambulatory settings as administered in intramuscular injections. However, currently circulating viral variants present less susceptibility to this drug than to other mAbs.

Although the main function sought in these mAbs is the ability to neutralize infection, there is growing evidence that the antibodies’ effector functions may play an important role in protection against the more severe forms of COVID-19. This protection would be related to the functions of cytotoxicity, phagocytosis and stimulation of the antibody-mediated cellular response [59]. For instance, the administration of sotrovimab in the COMET-ICE clinical trial on susceptible variants showed that a favorable outcome was associated with a decrease in SARS-CoV-2 viral load in the respiratory tract and the normalization of the expression of 10 of the genes significantly associated with the inflammatory response predictive of severe outcome in COVID-19 [60].

Drugs able to block the life cycle of SARS-CoV-2 have proved efficacious in controlling the progression of COVID-19 into more severe forms, particularly in individuals with risk factors. The spike protein on the surface of the SARS-CoV-2 membrane binds to cellular angiotensin-converting enzyme 2 (ACE2) in the host. For the fusion of viral and cellular membranes, and the entrance of the viral ribonucleoprotein complex into the cell, an interaction between the SARS-CoV-2 spike protein and the cellular transmembrane serine protease 2 (TMPRSS2) is needed [61]. In cases of viral infection via endocytosis, the cathepsin-L is responsible for releasing the viral RNA into the host cell cytoplasm. Cellular furins are also needed in the process of viral maturation [62]. Drugs that interfere with a list of host enzymes to block SARS-CoV-2 replication are still under study, and none of them are ready for clinical use. The main mechanism of action of these compounds under development is: (i) interference with ACE receptor; (ii) inhibition of TMPRSS2; (iii) inhibition of furins; (iv) inhibition of cathepsin-L; and (v) interference with viral endocytosis.

Antiviral drugs developed for the treatment of COVID-19 target viral proteins needed to complete SARS-CoV-2 replication in infected cells. According to the mechanism of action of the drug, these compounds may be classified as: (i) RNA-dependent RNA polymerase (RdRP) inhibitors; (ii) viral protease inhibitors and (iii) maturation inhibitors.

Three direct acting antivirals have gained approval for the treatment of COVID-19:

Remdesivir: This drug is an analog of adenosine, which inhibits viral replication by acting as an RdRP inhibitor when incorporated into the nascent RNA strands, causing the termination of RNA synthesis. The benefit of remdesivir among hospitalized patients taken together is not completely clear; however, in subjects with severe disease but not under ventilation, the use of this drug is associated with (i) accelerated recovery; (ii) lower need of mechanical ventilation and (iii) reduced mortality [63,64,65,66,67,68,69,70,71,72]. Although the benefit of remdesivir among ventilated patients is not fully clear, an observational study also proved a reduction in mortality with remdesivir in this subset of patients [73]. In hospitalized patients, remdesivir is given for 5 days, as longer treatment has not demonstrated additional benefit; longer durations may be considered in immunosuppressed patients as prolonged replication and selection of viral resistance is possible [74].

Remdesivir may be considered as a second-choice option in non-hospitalized patients with non-severe COVID-19 to prevent disease worsening, particularly in subjects that have not been vaccinated, or if other comorbidities are present. Antiviral treatment for three days is associated with a reduced need for visits to the hospital or hospitalization and lower mortality [46]. The intravenous route of administration of remdesivir may be viewed as a limitation for the use of this drug in the ambulatory setting; the oral formulation of the drug is currently under study.

Nirmatrelvir/ritonavir: This drug is an inhibitor of the viral 3CL protease, which breaks down the viral polyproteins into nonstructural proteins that are needed for the assembly of new viral particles. Nirmatrelvir/ritonavir has shown efficacy in ambulatory patients at risk of adverse outcome, including unvaccinated subjects or SARS-CoV-2-naïve subjects, in terms of providing shorter viral shedding and lower risk of hospitalization or death [48,75,76]. The efficacy of the drug is more pronounced in subjects with risk factors such as older age, immunosuppression and cardiovascular or neurological disease. The benefit of this antiviral drug in vaccinated subjects or in those with a history of COVID-19 without risk factors is doubtful. Nirmatrelvir/ritonavir has been authorized for treating mild-to-moderate COVID-19 in adults and children. The active drug is taken orally in combination with ritonavir, a potent cytochrome P450-3A4 inhibitor that improves the pharmacokinetic profile of the drug. The reappearance of some symptoms of COVID-19 within 10 days after the end of treatment with nirmatrelvir/ritonavir has been described in a small proportion of patients [75,77]. This clinical relapse is associated with viral rebound; that is, with positive SARS-CoV-2 antigen or RT-PCR tests. COVID-19 recurrence is, in general, self-limited, but in patients with a risk of severe immunosuppression, retreatment with antivirals may be considered.

Molnupinavir: Once the RdRp incorporates this ribonucleoside analog into nascent viral RNA, the replication process is altered by the accumulation of mutations, driving the virus to error catastrophe and making replication impossible. In ambulatory patients with mild-to-moderate COVID-19 with additional comorbidities, the drug has shown reductions in the risk of progression, but is not so clear in the need for hospitalization and mechanical ventilation, or in mortality [47,51,78,79,80,81]. According to comparative analysis, molnupiravir is considered less effective than remdesivir or nirmatrelvir/ritonavir, so it is only considered a marginal option for non-hospitalized patients at risk of progression. Furthermore, a recent evaluation by the EMA has recommended against the marketing and use of molnupiravir and called for the withdrawal of the drug, which has been recalled by the marketing company. A re-evaluation by the EMA with novel evidence is currently being undertaken [82].

Antiviral drugs are active only in the presence of viral replication, which is estimated to happen within 5 to 7 days from the onset of symptoms; the benefit of these drugs in patients with non-severe or severe COVID-19, as in those with risk factors for poorer outcomes including unvaccinated or virus-naïve subjects, may be extended to a longer period. Beyond 7 days from symptom onset, the indication of antivirals should not be based on real-time PCR results, as viral RNA may be detected days after viral replication has subsided. The cycle threshold (Ct) may be a surrogate marker of SARS-CoV-2 RNA concentration; higher mortality from COVID-19 has been shown in hospitalized patients with Ct values below 25 [83]. Subgenomic RNA detection and antigen tests also have a good correlation with active viral replication [84,85]. The presence of viral replication, rather than the need for oxygen supply, is for many the guide to indicate antivirals in hospitalized patients with COVID-19 [86].

## 3. Main Objective

The aim of this expert consensus (MODUS Project) is to generate recommendations to establish the patient profiles that can benefit most from the use of AVTM against SARS-CoV-2, so that with these interventions the prognosis of the infection is expected to improve. Consequently, these recommendations may also help avoid indiscriminate and futile prescribing of these drugs.

This review addresses the characteristics and risk factors among patients with the diagnosis of COVID-19 that indicate progression to severe forms with poor prognosis. These patients become candidates for AVTM to halt progression to severe COVID-19. This project does not simply aim to generate a list of risk factors, but rather to produce recommendations intended to promote therapeutic actions.

## 4. Methods

A review of the literature published between 1 July 2021 and 15 February 2022 was conducted to identify scientific publications (original articles, cohort and case–control studies, meta-analyses or systematic reviews) describing which risk factors are indicators of poor prognosis in patients with SARS-CoV-2 infection, making them candidates for the use of AVTM, understood as antivirals or monoclonal antibodies, to halt progression to severe forms of COVID-19. Ex-post, when deemed relevant by the experts, published evidence after 15 February 2022 was accepted for inclusion in the analysis to provide additional data (Figure 1).

The search was conducted in the following main sources of electronic reference libraries to access the available data: PubMed and The Cochrane Library. Related keywords, medical subject headings and free-text terms were used to search for the following concepts: COVID-19, SARS-CoV-2, poor prognosis, risk factors and mortality. The final search query was as follows: (((covid[Title/Abstract]) OR (sars-cov-2[Title/Abstract])) AND ((hospital admission[Title/Abstract]) OR (death[Title/Abstract]) or (mor*[title/abstract]) OR (bad outcomes[Title/Abstract]) OR (poor outcomes[Title/Abstract])) AND ((risk[Title/Abstract]) OR (predic*[Title/Abstract]) OR (prog*[Title/abstract]) OR (factor[Title/abstract])) AND (vacci*[Title/Abstract]))) AND ((“2021/6/1”[Date—Publication]: “3000”[Date—Publication])).

A total of 521 articles meeting the inclusion criteria were identified through the literature search. These records were analyzed according to title and abstract by 2 independent reviewers; if necessary, a third reviewer was involved in the decision to select or discard evidence. A total of 337 articles were discarded and 184 articles were selected. Once the evidence had been classified by headings, the articles were downloaded and forwarded to the editor responsible for each heading. The person responsible for each heading was tasked with distributing the bibliography among the team of assigned reviewers and with coordinating the review of the information and analysis of the evidence (GRADE). As mentioned above, in addition to the initial search provided to each writer, the authors were subsequently allowed, through a scoping review, to add additional references of high scientific value which, according to their expert judgement, provided new data that affected the quality and strength of the recommendations. A total of 159 articles outside the time range of the literature search were added.

A total of 278 articles were included in the final quantitative synthesis for the drafting of the list of risk factors and recommendations (Figure 2): 119 from the review of the literature and 159 from the additional independent review carried out by the actual editors a posteriori.

A total of 37 experts, selected by the coordinators of the project (FJC, PB and MS), participated in the development of this document; all work at Spanish medical centers and all have recognized experience in treating COVID-19, according to publications, communications or participation in scientific panels. The areas of knowledge that were covered by this group of experts were primary care, emergencies, internal medicine, geriatrics, pulmonology, hematology, oncology, rheumatology, infectious diseases, microbiology, transplant medicine, intensive care and public health. A total of 20 drafters read the scientific evidence and established the preliminary list of risk factors under each heading. A three-round consensus validation was carried out with a larger group of 37 experts to ratify what had been analyzed after the systematic review of the literature. The objective was to reach an expert consensus on recommendations to establish the patient profiles that may benefit most from the use of ATVM.

The three-round consensus was conducted in two distinct phases: a face-to face meeting with 26 experts divided into three groups (25 October 2022) and an online questionnaire answered by 34 experts to validate the recommendations. Consensus was regarded as having been reached when 80% of the experts ticked the highest degree of agreement on a scale of 1 to 9.

The final recommendations for the treatment of SARS-CoV-2 infection derived from this review should be taken into account while considering several limitations. The temporal frame of the studies included has left the most recent evidence without, particularly those studies that refer to vaccinated or immune populations or to infections caused by currently circulating viral variants. It may be that these factors, better immunity in the population and novel viral variants, have modified the specific influence of certain factors on the prognosis of COVID-19. To overcome these limitations, panelists have been able to consider studies carried out before and after vaccination was implemented. Additionally, in the review process, panelists were able to include updated relevant publications after February 2022 to assess the impact of the Omicron variant, the last to cause many infections worldwide.

Given that the evidence about the influence of risk factors on COVID-19 has been shifting along the pandemic, a Bayesian statistical analysis would have been the ideal approach for this review. The large amount of information gathered for consideration made this comparison impossible.

## 5. Results

Considering that the aim of the MODUS Project is to determine the clinical factors of poor prognosis and/or poor outcome and/or death in patients with SARS-CoV-2 infection, what follows is the final list of risk factors extracted following the review of the available evidence and the expert meetings. In Table 1, there is a summary of the risk factors that were considered by the panel, and the level of evidence and recommendation for this consideration. The wording of these profiles has been simplified as far as possible with the aim of generating a useful list of “patient at risk” profiles for clinical practice.

There are several considerations to be addressed: (i) these are the factors to be considered as of the diagnosis of SARS-CoV-2 infection and at any level of care; (ii) these factors, unless otherwise specified, are considered to be applicable to the entire population meeting the criteria, regardless of whether the patient has received any dose(s) of vaccine, the time since the last dose and/or whether the patient has had COVID-19 and the time elapsed since then; and (iii) it also details the factors which, based on the available evidence, cannot be considered as risk factors, and nor can actions different from those that would be applied in the general population be recommended.

It is important to understand that the purpose of identifying several risk factors is to promote therapeutic action to prevent progression to more severe stages of the disease; in this case, to indicate specific treatment against SARS-CoV-2 with monoclonal antibodies (mAb) and/or antivirals within the first few days after diagnosis and for anyone presenting any of the factors listed (provided there is no other medical contraindication).

For each risk factor, the quality of the evidence supporting its consideration as a key factor is indicated, as is the strength of the recommendation for therapeutic action according to the criteria of the multidisciplinary group of experts participating in the MODUS Project.


**
Elderly, frail and institutionalized patients
**


***Rationale***. Age is one of the main prognostic factors of a poor clinical outcome (hospitalization, need for intensive care or death) in patients infected with SARS-CoV-2, with a strong association with the age of 65 years and a maximum from the age of 70 years onwards, both in unvaccinated and vaccinated populations [87,88,89,90,91,92,93,94,95,96,97,98,99,100,101,102]. There is also a correlation between comorbidity and poorer clinical outcome, including mortality, at all ages, although particularly after 60 years of age. The strength of this association increases with the number of comorbidities affecting the patient, with the association being highest for three or more pathological processes, both in unvaccinated and vaccinated subjects, including pediatric patients [9,40,87,88,89,90,91,92,93,94,95,96,97,98,99,100,101,102,103,104,105,106]. In nursing homes for older adults (NHOA), the factors associated with poorer clinical outcomes are age, health status and frailty [31,88,90,92,95,107,108,109,110,111,112,113]. There are different scales for determining frailty (CFS, Rockwood, FRAIL, ISAR, Green) [114], although in all of them the worst prognosis is defined by a score of 4 or higher. Table 2 summarizes the final recommendations of the panel.


**
Body weight
**


***Rationale.*** In addition to the negative effect that obesity has on its own, particularly when BMI is around 40 kg/m^2^, obesity is often associated with other comorbidities such as diabetes or cardiovascular disease, and has an impact on the prognosis of patients with SARS-CoV-2 infection, both in the unvaccinated and vaccinated population, including the pediatric setting [97,98,103,107]. Table 3 summarizes the final recommendations of the panel.


**
Kidney function
**


***Rationale.*** Renal failure is an independent risk factor for mortality in any clinical process, both infectious and non-infectious. The impact of the association of functional renal failure (CrCl < 50 mL/min) with or without plasma exchange on the prognosis of patients with SARS-CoV-2 infection has been reviewed in both unvaccinated and vaccinated populations, including the pediatric population [9,38,87,107,115]. The prognosis in these patients, in terms of ICU admission and death, increases significantly with age (over 70 years), with comorbidity (CCI of 2 or more) and with the intensity of renal function deterioration (CrCl < 40 mL/min and hemodialyzed) [9,38,87,107,115]. Table 4 summarizes the final recommendations of the panel.


**
Liver function
**


***Rationale.*** The prognostic impact of liver failure has only been demonstrated in the most advanced stages. The impact of liver failure on the prognosis of patients with SARS-CoV-2 infection has been reviewed in both unvaccinated and vaccinated populations, including pediatrics [88,89,107,116]. Table 5 summarizes the final recommendations of the panel.


**
Solid organ transplantation
**


***Rationale.*** The impact of solid organ transplantation (SOT) on the clinical course of SARS-CoV-2 infection has been analyzed [115,116,117,118,119]. A review based on 15 articles published up until March 2021 (with 265,839 participants, including 1485 recipients) showed that SOT was associated with an increased risk of ICU admission compared to non-transplant recipients (OR: 1.57; *p* = 0.02). The SOT population also had a higher adjusted mortality (HR: 1.54; *p* = 0.037) [117]. The data also show that the presence of comorbidities is more frequent in SOT recipients with SARS-CoV-2 infection compared to recipients without infection. In turn, SARS-CoV-2 infection increases the risk of developing post-transplant complications, such as major cardiovascular events, acute renal failure or graft rejection [115].

In a large multicenter cohort of SOT recipients during the first wave of the pandemic, the need for hospitalization and invasive mechanical ventilation (78% and 31%, respectively) was high, as was 28-day mortality (20.5%). Age over 65 years and the presence of congestive heart failure, chronic lung disease and obesity have been identified as independent risk factors for mortality or a poor outcome [118,119]. It is also worth noting that mortality in SOT recipients requiring hospitalization did not decrease significantly during the second wave of the pandemic [116].

SARS-CoV-2 vaccines, based either on non-replicating viral vectors or messenger RNA technology, show a lower efficacy in SOT recipients compared to the general population in terms of seroconversion rate or neutralizing antibody titers [120,121,122]. As a consequence, the development of severe SARS-CoV-2 infection requiring hospitalization has been reported in SOT recipients who had received a complete vaccination regimen [123,124]. Mortality in these cases of breakthrough infection can be considerable (9.3%) [124].

The administration of monoclonal antibodies directed against the spike glycoprotein in SOT recipients with SARS-CoV-2 infection has been shown to be safe and effective in reducing the risk of progression to severe disease, although the available evidence comes from observational studies [125,126,127]. More specifically, the use of sotrovimab in kidney transplant recipients over 55 years or with other risk factors for severe infection (diabetes, obesity, graft dysfunction, coronary heart disease or chronic lung disease) infected with the SARS-CoV-2 Omicron variant reduced both mortality and the need for ICU admission compared to a historical control group not treated with sotrovimab [127].

In the light of the data analyzed, it may be concluded that SOT is a prognostic factor for a poor clinical outcome (need for hospitalization, ICU admission or death) in SARS-CoV-2 infection, with the risk being higher in older recipients with graft dysfunction (particularly renal) or comorbidities. The association applies to both vaccinated and unvaccinated populations, although the risk of a poor outcome in the latter is lower [107]. Table 6 summarizes the final recommendations of the panel.


**
SARS-CoV-2 PCR-positive graft donors or recipients
**


***Rationale.*** The potential feasibility of deceased COVID-19 patients with a diagnostic test for active SARS-CoV-2 infection (usually PCR) being able to act as solid organ or hematopoietic progenitor donors was considered from the outset of the pandemic. There are theoretical arguments for the risk of transmission of infection via solid organs and, to a lesser extent, hematopoietic progenitors; however, the available evidence for viable virus isolation in non-pulmonary organs and tissues is inconsistent [128].

Several case series, both multi- [129,130] and single-center [131,132,133] and individual cases [134], have shown favorable results with the use of organs (other than the lung) from deceased donors with a positive PCR for SARS-CoV-2 at the time of donation or in the days immediately preceding donation [135]. None of these studies demonstrated organ-mediated transmission or a worse-than-expected graft outcome considering donor and recipient characteristics [129,130,131,132,133,134].

In some centers, and in a pre-vaccination context, priority was given to selecting recipients with a positive serology for SARS-CoV-2 on the waiting list in order to ensure the presence of natural immunity [133]. It should be noted that none of these articles considered the preventive administration of antiviral drugs or monoclonal antibodies in the donor with the specific aim of reducing the theoretical risk of transmission, although it is true that there is no evidence to suggest that the administration of any type of preventive treatment could have a discernible impact on this theoretical risk of transmission.

In fact, the only documented cases of via-graft SARS-CoV-2 transmission occurred in lung transplant recipients and shared the common feature that no pre-donation molecular testing (PCR) had been performed on a donor lower respiratory tract sample [136,137]. Therefore, according to the evidence reviewed, the use of solid organs (other than lung and intestine) and hematopoietic progenitors from donors diagnosed with an active SARS-CoV-2 infection (PCR-positive) has been shown to be safe. Table 7 summarizes the final recommendations of the panel.


**
Hematopoietic stem cell transplantation
**


***Rationale.*** Several observational studies, both retrospective [9,138,139,140] and prospective [141], have analyzed the clinical course of SARS-CoV-2 infection in hematopoietic stem cell transplantation (HSCT) recipients.

In a large multicenter cohort (n = 318) published in the pre-vaccine period, an ICU admission rate of 14% was observed, with an overall 30-day survival of 68% [15]. Being older [15,141], male gender [15], a period of less than one year since autologous hematopoietic stem cell transplantation (AHSCT), the presence of comorbidities and the presence of a high risk of death [15,140], the presence of comorbidities [138], the degree of immunosuppression [140,141], neutropenia [138] and certain clinical features at the time of diagnosis, such as the presence of pneumonia [138], were identified as predictors of death or unfavorable outcome. In addition, no significant prognostic differences were observed between autologous and allogeneic procedures [15,141].

On the other hand, none of the studies reviewed, including a systematic review and meta-analysis [32], reported the administration of monoclonal antibodies or antiviral drugs against SARS-CoV-2 and neither were specific recommendations made in two clinical practice guidelines [142,143].

In conclusion, and according to the evidence analyzed, HSCT is a prognostic factor for a poor clinical outcome (need for hospitalization, ICU admission or death) in patients infected with SARS-CoV-2. Table 8 summarizes the final recommendations of the panel.


**
Oncologic patients
**


***Rationale.*** Oncology and oncohematology patients have higher morbidity and mortality due to COVID-19. The data published to date corroborate this, while also shedding some light on the possibility of identifying subgroups of oncology patients who have a particularly poorer prognosis.

The existing evidence in this regard is scant and comes mostly from observational studies with a very heterogeneous data collection, obvious limitations considering the complexity of the moment and the imperative need for real-life data.

Moreover, the analysis of the evidence suggests that patients with active cancer have a higher risk of mortality from SARS-CoV-2 than the general population [36,90,106,107,108,110,144,145,146,147,148,149]. There are certain factors associated with these patients that may help to identify subgroups with a higher risk of increased mortality, including: (i) factors associated with neoplastic pathology (the presence of hematological malignancies) [110,146,147,148,150]; (ii) the presence of active disease and/or being on treatment [36,108,110,144,146,147,151,152,153,154,155,156]; (iii) refractory disease [156]; (iv) some subtypes of hematological malignancies (chronic lymphoid leukemia) [150] and solid tumors (lung cancer) [108,144]; (v) treatment-dependent factors (treatment with antiCD20) [156]; (vi) patient-dependent factors such as advanced age [88,90,144,146,150,151,156], presence of comorbidities [36,151,152] and high BMI [144]; and (vii) lower serological response to vaccination [153,154,155]. The evidence reviewed leads to the conclusion that oncology/oncohematology patients are particularly vulnerable to severe forms of SARS-CoV-2.

On the other hand, in the analysis of the evidence for these subgroups, patients with cancer in remission and/or without active treatment appear to have a lower mortality than those on active treatment [36,108,110,144,146,147,151]. Furthermore, these patients do not present a higher risk of COVID-19 mortality than the general population.

Patients with lymphoid hematological diseases—acute lymphoid leukemia (ALL), chronic lymphocytic leukemia (CLL) and/or non-Hodgkin’s lymphoma (NHL)—due to their prolonged immunosuppressed state, are more susceptible to SARS-CoV-2 infection compared to patients in remission and/or without active treatment, although there is no difference in mortality between them according to the underlying disease status [148,150,152].

However, the data do show that patients over 12 years of age weighing at least 40 kg and with stable-phase cancer and one or more comorbidities are, like the general population, at increased risk of severe forms of SARS-CoV-2 infection over and above their cancer patient status. These comorbidities include [88,145,157] (i) stable structural or functional airway disease, without oxygen dependency at the time of diagnosis; (ii) COPD on oxygen therapy with no change in oxygen dependence; (iii) functional renal insufficiency (CrCl < 50 mL/min) with or without plasma exchange; (iv) obesity with BMI > 40 as a sole disease, or BMI >30 and any other comorbidity; (v) insulin-dependent diabetes mellitus with poor control (glycated hemoglobin (Hb-A1c) > 6.5%); (vi) functional liver failure (Child–Pugh ≥ 10 points); (vii) established ischemic heart disease, with myocardial dysfunction or with normal myocardial function and two or more cardiovascular risk factors (hypertension (HT), hyperlipidemia, diabetes, obesity); (viii) congestive heart failure with a New York Heart Association (NYHA) classification ≥II and (ix) hematopoiesis disorders.

In summary, oncology/oncohematology patients with stable-phase cancer are not at increased risk of developing more severe forms of COVID-19 due to the nature of their pathology as long as they do not have associated comorbidities—which are already a risk factor for the general population—or are in a state of prolonged immunosuppression due to treatment. Table 9 summarizes the final recommendations of the panel.


**
HIV infection
**


***Rationale.*** Studies comparing the clinical features and prognosis of COVID-19 in people with and without human immunodeficiency virus (HIV) infection have yielded conflicting results. Some have found no significant differences in these aspects between the two populations [158,159,160,161,162,163,164,165,166,167,168,169], whereas others have found a higher risk of hospitalization, ICU admission or death from COVID-19 in people with HIV than in the general population [170,171,172,173,174,175]. These results must be viewed in the light of the different levels of antiretroviral treatment (ART) coverage and differences in socio-economic status and living conditions between people with and without HIV across the geographic regions in which these studies have been conducted.

In any case, many observational studies indicate that the prognosis of COVID-19 in people with HIV is largely determined by the same factors that influence the general population, such as gender, age and the presence of comorbidities [33,165,172,174,175,176,177,178,179,180,181,182,183,184,185,186,187], and by socio-economic status, living conditions and ethnicity [166,174,175,179,182,188]. The importance of demographic characteristics and comorbidity in this population is reinforced by the absence of prognostic differences between people with and without HIV in cohort studies matched for these variables [170,180,189,190].

The effect of the factors directly related to HIV on the prognosis of COVID-19 is still unknown, especially since many of the studies that have examined these aspects have been conducted in countries with good ART coverage and a low proportion of patients with immunosuppression or detectable viraemia. Nevertheless, an association between low CD4+ T-cell counts and increased risk of hospitalization, ICU admission and mortality from COVID-19 has been observed [33,107,170,187,188,191,192], with a cut-off point for defining a low CD4+ T-cell count of <200 cells/mm^3^ having been established in some studies [33,167,189] and of <350 cells/mm^3^ in others [187,191,193]. A poorer prognosis for COVID-19 has also been observed in HIV patients with a low CD4+ T-cell count nadir [178,187,194], with a threshold CD4+ T-cell count of <200 cells/mm^3^ established in one of them [187].

Detectable viral load has also been identified as a poor prognostic factor in several studies [163,183,191,192,195]. Among the available evidence, a large observational study in the US found no such association, although most HIV patients were on ART and had suppressed viral load [187]. Only in one retrospective study with fewer than 100 hospitalized patients in New York City was the risk of intubation and death higher in patients with a detectable viral load [195].

Finally, the effect of antiretroviral drugs on the natural history of COVID-19 is a highly controversial area. Some studies have found an association between treatment with tenofovir disoproxil fumarate (TDF) and emtricitabine (FTC) with a lower risk of SARS-CoV-2 infection and a better course of COVID-19 [167,186]. However, other studies have found no association between any antiretroviral and severity of infection [172,176,178,179]. Table 10 summarizes the final recommendations of the panel.


**
Primary and other secondary immunodeficiencies
**


***Rationale.*** The pathogenesis of severe COVID-19 may be related to the characteristics of the infection, the impairment of the immune response leading to viral persistence and the effects of an excessive inflammatory response. In the first two cases, antiviral therapies are warranted, and in the second case, the use of immunomodulators for the treatment of severe COVID-19 is warranted.

Inborn errors of immunity (IEIs) are rare genetic defects that result in reduced expression or reduced or increased activity of certain proteins. These genetic changes affect germ cells and predispose to usually severe, even fatal, immune dysfunctions. These dysfunctions can lead to allergic reactions, recurrent opportunistic infections, autoimmune phenomena or the development of tumors. More specifically, with regard to acute viral infections, increased susceptibility to enterovirus, herpes simplex virus type I or Epstein–Barr virus, respiratory infections and reaction to attenuated vaccines have been reported [196,197,198,199,200,201].

More than 450 different IEIs are known, varying widely in their presentations and clinical manifestations; a distinction may be made between IEIs with impaired innate (e.g., chronic granulomatous disease) or adaptive (e.g., X-linked agammaglobulinemia, common variable immunodeficiency, etc.) immunity. In other cases, immune dysfunction involves molecular pathways, such as the complement pathway, IFNϓ/IL-12, IL-17, etc.

There are data on the evolution of COVID-19 in individuals affected by defects of innate immunity, humoral immune dysfunction (including common variable immunodeficiency (CVID), Bruton’s tyrosine kinase, agammaglobulinemia and hypogammaglobulinemia), defects in phagocytosis or bone marrow failure (including chronic granulomatous disease), immune dysregulation (including polyglandular autoimmune syndrome type I (PAS-1)), autoinflammatory processes (including familial Mediterranean fever and Aicardi–Goutières Syndrome) or complement deficiency (including hereditary angioedema or C3 deficiency).

The studies conducted in cohorts of patients with IEIs have not found a significantly different frequency of severe COVID-19 to that of the general population [202,203], although it is true that there is a much higher increase in recurrent/persistent forms of COVID-19 in real-life data, necessitating repeated treatment of patients with antivirals in combination with monoclonal antibodies (previously with plasma from convalescent donors) [204].

An estimated 10% of patients with IEIs have asymptomatic COVID-19 and 50% have mild COVID-19. However, severe cases occur at younger ages than expected in the general population and the ICU admission rate is also higher [205]. Globally, for patients with IEIs, the case fatality rate is between 8.1% and 9.5% [203,205], which could be compared to the 5–18% case fatality rate for the general population in the period of the pandemic when these studies were conducted. Thus, in a worst-case scenario, the COVID-19 case fatality rate in patients with IEIs is twice that of the general population.

On the other hand, in general, no differences have been observed between the different types of IEIs and severity of COVID-19 [205]. The poorer prognosis of COVID-19 in people with IEIs seems rather to depend on the coexistence of other comorbidities. This is the case for CVID, the most frequent immune disorder, which involves defective humoral immune function and various alterations in the cellular compartment.

CVID causes recurrent and severe respiratory infections which, in about 30% of patients, leave non-infectious complications (bronchiectasis, asthma and COPD; it is also associated with autoimmune and inflammatory phenomena leading to interstitial lung disease (ILD)). Patients with lung disease have a moderate risk of severe disease due to COVID-19 with hospitalization and possibly ICU admission and a moderate risk of death due to COVID-19. Of these comorbidities, COPD and ILD aggravate the prognosis of COVID-19, while the presence of bronchiectasis or asthma does not seem to have such a negative effect [206,207,208].

Peculiarly, APS-1 causes a higher risk of complications due to COVID-19. In this case, the risk of hospitalization (73%), ICU admission (58%) and death (15%) due to COVID-19 is much higher than in other IEIs [203,209,210,211]. The immune dysfunction associated with APS-1 is the production of autoantibodies against IFN-1. This mechanism of immunosuppression is more frequent in males and older people [212,213]. It has been proposed that the detection of these autoantibodies is a COVID-19-independent predictor of poor prognosis [214]. In X-linked Toll-like receptor 7 (TLR7) deficiency, and thus exclusively affecting males, an increased risk of severe or fatal COVID-19 has also been found [215,216,217].

On the other hand, there is little evidence of the risk of a poor clinical outcome in patients with anatomical or functional asplenia. There is also a predisposition to complications [21,218].

Finally, it is important to mention that activation of the cytokine storm and complement cascade is associated with severe COVID-19 [217,219]. For this reason, people with C3 deficiency or hereditary angioedema with C1 inhibitor deficiency have often been seen to have asymptomatic or mild COVID-19 [220,221]. Table 11 summarizes the final recommendations of the panel.


**
Immune-mediated inflammatory diseases
**


***Rationale.*** Patients with immune-mediated inflammatory diseases (IMIDs) may be at increased risk of severe COVID-19 because of the immune dysfunction caused by the actual disease or because of the immunosuppressive effect of many of the drugs used to treat these chronic conditions [222].

The evidence available regarding the possible negative effects of IMID comorbidity on the evolution of COVID-19 is contradictory. Some studies find a higher risk of hospitalization for COVID-19 in patients with Rheumatic and Musculoskeletal Diseases (RMSDs), while others indicate that they have a similar risk to the general population [223,224,225,226,227,228,229,230,231,232,233,234,235,236] or even lower [222]. These differences may well occur because in many studies it is not possible to separate the effect of the disease from that of the immunosuppressive treatment. On the other hand, the likelihood of hospitalization for causes other than COVID-19 may also be increased in patients with IMIDs. According to some of these studies, it may be said that the risk of death from COVID-19 is not increased by having IMID [222,225,236,237]. It is not possible to distinguish between the different IMIDs and their effect on the prognosis of COVID-19.

With regard to the effect of the immunosuppressive drugs used to treat MRDs, there is moderate evidence that glucocorticoids increase the severity of COVID-19, particularly if the chronic doses used exceed 10 mg daily [237,238,239,240,241,242,243,244,245].

There is no evidence of an association of the use of methotrexate, leflunomide or biological agents (anti-TNF, anti-IL1, anti-IL6, anti-IL17, anti-IL 12/23, anti-IL23, abatacept or belimumab) with severity or mortality consistently across all published studies. On the other hand, the data on JAK kinase inhibitors are inconsistent, as some studies show an association with severity or mortality, whereas others do not detect this effect.

It should also be noted that, in the first waves of the COVID-19 pandemic, hyperinflammatory states of severe cases hospitalized with SARS-CoV-2 infection were treated with IL-1 blocking agents (anakinra) and JAK inhibitors (e.g., baricitinib), and subsequent clinical trials demonstrated the protective and effective value of these drugs in counteracting the complications of cytokine release. Similarly, after some controversy, the same occurred with the indication of tocilizumab (IL-6 inhibitor agent) in severe cases of COVID-19.

Therefore, these drugs could play a certain protective role at the appropriate time in the evolution of patients with severe SARS-CoV-2 infection and, with regard to patients already receiving them as a base treatment for their IMID, it is possible that, while they cannot demonstrate a preventive effect by attenuating the baseline inflammatory state and secondary to COVID-19, at least they might not be associated with increased disease severity.

There are fewer studies of other immunosuppressants, and they are inconsistent. One study shows an association with mortality for mycophenolate, azathioprine, calcineurin inhibitors and cyclophosphamide together and another for mycophenolate individually. However, other studies do not detect this association with mycophenolate [233].

In contrast, there are consistent data on anti-CD20 drugs (rituximab and ocrelizumab), associating the latter with greater severity and mortality due to COVID-19 and lower vaccine protection on account of insufficient or no response to the vaccines administered, at least up to those comprising the 1st-, 2nd- and 3rd-dose strategy, although this probably extends to any vaccination strategy, including boosters [237,242,246,247,248,249,250].

After the consensus rounds, not included in recommendations, the expert group found new evidence on one of the points of lower consensus due to controversies in the published data. This new evidence supports a higher risk of progression to severe forms of COVID-19 disease in patients with RMSDs compared to the general population, and could support the need to promote therapeutic actions in these patients to prevent progression [251]. Table 12 summarizes the recommendations of the panel.

## Figures and Tables

**Figure 1 viruses-15-01449-f001:**
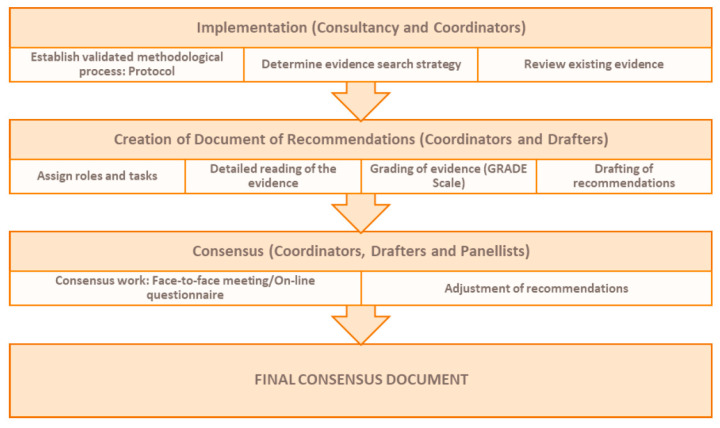
Review process followed in the MODUS project.

**Figure 2 viruses-15-01449-f002:**
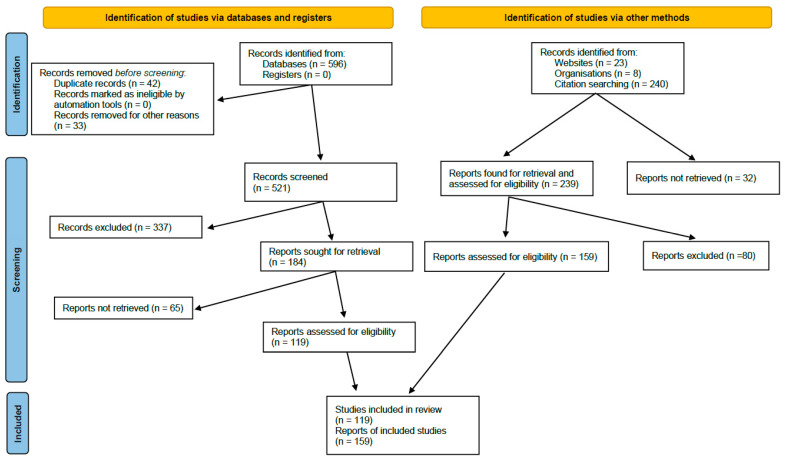
PRISMA diagram of the literature review followed in the MODUS project.

**Table 1 viruses-15-01449-t001:** List of risk factors.

Profile of Patients with an Increased Risk Factor for Severe Forms of COVID-19	Quality of the Evidence	Strength of the Recommendation * (Therapeutic Action to Prevent Progression to More Severe Stages of the Disease)
Age, frailty and institutionalized persons		
Being a patient over 80 years of age, regardless of vaccination status or history of SARS-CoV-2 infection	HIGH	STRONG—VERY STRONG
Being a patient over 65 years of age, regardless of vaccination status or history of SARS-CoV-2 infection, with three or more chronic-risk diseases **	HIGH	STRONG
Being a patient over 65 years of age with a chronic-risk disease ** and without vaccination or previous infection ***	HIGH	STRONG
Being a person residing in a nursing home, regardless of vaccination status or history of SARS-CoV-2 infection	MODERATE	MODERATE
Being a person with moderate–severe frailty (>4 on the Clinical Frailty Scale), regardless of vaccination status or history of SARS-CoV-2 infection	MODERATE	MODERATE
**Obesity**		
Being a patient with a BMI >35 and without vaccination or previous infection ***	LOW	MODERATE
**Renal failure**		
Being a patient on replacement therapy (hemodialysis/peritoneal dialysis) for chronic kidney disease, regardless of vaccination status or history of SARS-CoV-2 infection	LOW	MODERATE
Being a patient with chronic kidney disease and glomerular filtration rate < 30 mL/min, regardless of vaccination status or history of SARS-CoV-2 infection	LOW	MODERATE
**Liver failure**		
Being a patient with liver failure in cirrhosis or liver failure (Child–Pugh B or C, ≥7 points) and without vaccination or previous immunity ***	LOW	MODERATE
**Solid organ transplantation**		
Being a solid organ transplant recipient, regardless of vaccination status or history of SARS-CoV-2 infection	LOW	MODERATE
**PCR-positive hematological or solid organ donors/recipients**		
Being a solid organ or hematopoietic progenitor donor with a positive PCR for SARS-CoV-2 is NOT regarded as an increased risk factor for SARS-CoV-2 transmission	LOW	MODERATE
**Hematopoietic Stem Cell Transplantation (HSCT)**		
Being a hematopoietic stem cell transplant (HSCT) recipient, regardless of vaccination status or history of SARS-CoV-2 infection	LOW	MODERATE
**Oncology/oncohematology patient**		
Being an oncology patient undergoing antineoplastic chemotherapy, regardless of vaccination status or history of SARS-CoV-2 infection	MODERATE—HIGH	STRONG
Being an oncohematology patient with active malignancy, regardless of vaccination status or history of SARS-CoV-2 infection	MODERATE—HIGH	STRONG
**Poorly controlled HIV infection with one or more comorbidities**		
Being an HIV-infected patient with a CD4+ cell count < 200/mm^3^, regardless of vaccination status or history of SARS-CoV-2 infection	MODERATE	STRONG
Being an HIV-infected patient with a CD4+ cell count of 200–350/mm^3^, regardless of vaccination status or history of SARS-CoV-2 infection	MODERATE	MODERATE
Being a patient with HIV infection and poorly maintained virological control of HIV, regardless of vaccination status or history of SARS-CoV-2 infection	VERY LOW	WEAK
Being a patient with HIV infection and CD4+ cell count > 350/mm^3^ is NOT considered an increased risk factor	LOW	WEAK
Being a patient with HIV infection and having had a CD4+ nadir <200/mm^3^ is NOT considered an increased risk factor	HIGH	STRONG
**Poorly controlled primary or secondary immunodeficiencies**		
Being a patient with primary congenital immunodeficiency, regardless of vaccination status or history of SARS-CoV-2 infection	LOW	WEAK
Being a patient with functional or anatomical asplenia, regardless of vaccination status or history of SARS-CoV-2 infection	LOW	WEAK
**Chronic steroid therapy and immune-mediated diseases**		
Being a patient with certain systemic or immune-mediated autoimmune diseases ****, depending on stage and treatment modality, regardless of vaccination status or history of SARS-CoV-2 infection	LOW	MODERATE
Being a patient on chronic glucocorticoid therapy, regardless of vaccination status or history of SARS-CoV-2 infection	MODERATE	MODERATE
Being a patient on anti-CD20 treatment, regardless of vaccination status or history of SARS-CoV-2 infection	HIGH	STRONG
There is NOT enough published evidence to regard patients on treatment with immunosuppressants other than those previously indicated (e.g., JAK or TNF inhibitors) as an increased risk factor	LOW	STRONG(as there is no evidence to recommend its use)

* A cut-off point for reaching consensus was established when 80% of the experts ticked maximum agreement (on a scale of 1 to 9: score ≥ 7). ** Pulmonary and/or cardiovascular diseases (such as COPD, severe asthma, adult cystic fibrosis, interstitial lung disease, history of pulmonary hypertension, pulmonary thromboembolism or tuberculosis, etc.), chronic kidney or liver disease and certain degenerative neurological diseases (e.g., Down’s syndrome, dementia). *** Complete vaccination scheme including booster dose within the last 6 months and/or less than 5 months after diagnosis of infection. **** In addition to a diagnosis of IMID, it may also be influenced by the type of immune-mediated inflammatory disease per se, its stage or degree of involvement and extent, especially additional comorbidities, and above all, the type of disease-modifying treatment, biological or non-biological, and the family to which the latter belong, as well as their target and mechanism of action.

**Table 2 viruses-15-01449-t002:** Therapeutic recommendations for patients with COVID-19 according to greater age, frailty and institutionalization.

Profile of Patients with an Increased Risk Factor for Severe Forms of COVID-19	Quality of the Evidence	Strength of the Recommendation (Therapeutic Action to Prevent Progression to More Severe Stages of the Disease)	Meeting (n = 13) *	Online Consensus (n = 33) *
Being a patient over 80 years of age, regardless of vaccination status or history of SARS-CoV-2 infection	HIGH	STRONG–VERY STRONG RECOMMENDATION	100%	100%
Being a patient over 65 years of age, regardless of vaccination status or history of SARS-CoV-2 infection, with three or more chronic-risk diseases **	HIGH	STRONG	100%	100%
Being a patient over 65 years of age with a chronic-risk disease ** and without vaccination or previous infection ***	HIGH	STRONG	100%	100%
Being a person residing in a nursing home, regardless of vaccination status or history of SARS-CoV-2 infection	MODERATE	MODERATE	100%	90.90%
Being a person with moderate–severe frailty (>4 on the Clinical Frailty Scale), regardless of vaccination status or history of SARS-CoV-2 infection	MODERATE	MODERATE	100%	90.90%

* A cut-off point for reaching consensus was established when 80% of the experts ticked maximum agreement (on a scale of 1 to 9: score ≥ 7). ** Pulmonary and/or cardiovascular diseases (such as COPD, severe asthma, adult cystic fibrosis, interstitial lung disease, history of pulmonary hypertension, pulmonary thromboembolism or tuberculosis, etc.), chronic kidney or liver disease and certain degenerative neurological diseases (e.g., Down’s syndrome, dementia). *** Complete vaccination schedule including booster doses within the last 6 months and/or less than 5 months after diagnosis of infection.

**Table 3 viruses-15-01449-t003:** Therapeutic recommendations for patients with COVID-19 according to elevated body weight.

Profile of Patients with an Increased Risk Factor for Severe Forms of COVID-19	Quality of the Evidence	Strength of the Recommendation (Therapeutic Action to Prevent Progression to More Severe Disease)	Meeting (n = 13) *	Online Consensus (n = 33) *
Being a patient with a BMI > 35 and without vaccination or previous infection **	LOW	MODERATE	100%	100%

* A cut-off point for reaching consensus was established when 80% of the experts ticked maximum agreement (on a scale of 1 to 9: score ≥ 7). ** Complete vaccination schedule including booster dose within the last 6 months and/or less than 5 months after diagnosis of infection.

**Table 4 viruses-15-01449-t004:** Therapeutic recommendations for patients with COVID-19 according to impairment in kidney function.

Profile of Patients with an Increased Risk Factor for Severe Forms of COVID-19	Quality of the Evidence	Strength of the Recommendation (Therapeutic Action to Prevent Progression to More Severe Stages of the Disease)	Meeting (n = 13) *	Online Consensus (n = 33) *
Being a patient on replacement therapy (hemodialysis/peritoneal dialysis) for chronic kidney disease, regardless of vaccination status or history of SARS-CoV-2 infection	LOW	MODERATE	100%	100%
Being a patient with chronic kidney disease and glomerular filtration rate < 30 mL/min, regardless of vaccination status or history of SARS-CoV-2 infection	LOW	MODERATE	100%	96.96%

* The cut-off point for reaching consensus was established when 80% of the experts ticked maximum agreement (on a scale of 1 to 9: score ≥ 7).

**Table 5 viruses-15-01449-t005:** Therapeutic recommendations for patients with COVID-19 according to presence of liver failure.

Profile of Patients with an Increased Risk Factor for Severe Forms of COVID-19	Quality of the Evidence	Strength of the Recommendation (Therapeutic Action to Prevent Progression to More Severe Stages of the Disease)	Meeting (n = 13) *	Online Consensus (n = 33) *
Being a patient with liver failure in a state of cirrhosis or liver failure (Child–Pugh B or C, ≥7 points) and without vaccination or previous immunity **	LOW	MODERATE	100%	96.96%

* The cut-off point for reaching consensus was established when 80% of the experts ticked maximum agreement (on a scale of 1 to 9: score ≥ 7). ** Complete vaccination schedule including booster dose within the last 6 months and/or less than 5 months after diagnosis of infection.

**Table 6 viruses-15-01449-t006:** Therapeutic recommendations for patients with COVID-19 in solid organ transplant recipients.

Profile of Patients with an Increased Risk Factor for Severe Forms of COVID-19	Quality of the Evidence	Strength of the Recommendation (Therapeutic Action to Prevent Progression to More Severe Stages of the Disease)	Meeting (n = 6) *	Online Consensus (n = 33) *
Being a solid organ transplant recipient, regardless of vaccination status or history of SARS-CoV-2 infection	LOW	MODERATE	100%	100%

* The cut-off point for reaching consensus was established when 80% of the experts ticked maximum agreement (on a scale of 1 to 9: score ≥ 7).

**Table 7 viruses-15-01449-t007:** Therapeutic recommendations for patients with COVID-19 as graft donors or recipients.

Profile of Patients with an Increased Risk Factor for Severe Forms of COVID-19	Quality of the Evidence	Strength of the Recommendation (Therapeutic Action to Prevent Progression to More Severe Stages of Disease)	Meeting (n = 6) *	Online Consensus (n = 33) *
Being a solid organ or hematopoietic progenitor donor with a positive PCR for SARS-CoV-2 is NOT considered an increased risk factor for SARS-CoV-2 transmission	LOW	MODERATE	100%	90.90%

* The cut-off point for reaching consensus was established when 80% of the experts ticked maximum agreement (on a scale of 1 to 9: score ≥ 7).

**Table 8 viruses-15-01449-t008:** Therapeutic recommendations for patients with COVID-19 with hematopoietic stem cell transplant.

Profile of Patients with an Increased Risk Factor for Severe Forms of COVID-19	Quality of the Evidence	Strength of the Recommendation (Therapeutic Action to Prevent Progression to More Severe Stages of the Disease)	Meeting (n = 6) *	Online Consensus (n = 33) *
Being a hematopoietic stem cell transplant (HSCT) recipient, regardless of vaccination status or history of SARS-CoV-2 infection	LOW	MODERATE	100%	100%

* A cut-off point for reaching consensus was established when 80% of the experts ticked maximum agreement (on a scale of 1 to 9: score ≥ 7).

**Table 9 viruses-15-01449-t009:** Therapeutic recommendations for oncologic and oncohematologic patients with COVID-19.

Profile of Patients with an Increased Risk Factor for Severe Forms of COVID-19	Quality of the Evidence	Strength of the Recommendation (Therapeutic Action to Prevent Progression to More Severe Stages of Disease)	Meeting (n = 6) *	Online Consensus (n = 33) *
Being an oncology patient undergoing antineoplastic chemotherapy, regardless of vaccination status or history of SARS-CoV-2 infection	MODERATE	STRONG	100%	100%
Being an oncohematology patient with active malignancy, regardless of vaccination status or history of SARS-CoV-2 infection	MODERATE	STRONG	100%	96.96%

* The cut-off point for reaching consensus was established when 80% of the experts ticked maximum agreement (on a scale of 1 to 9: score ≥ 7).

**Table 10 viruses-15-01449-t010:** Therapeutic recommendations for HIV-infected patients with COVID-19.

Profile of Patients with an Increased Risk Factor for Severe Forms of COVID-19	Quality of the Evidence	Strength of the Recommendation (Therapeutic Action to Prevent Progression to More Severe Stages of the Disease)	Meeting (n = 6) *	Online Consensus (n = 33) *
Being an HIV-infected patient with a CD4+ cell count < 200/mm^3^, regardless of vaccination status or history of SARS-CoV-2 infection	MODERATE	STRONG	100%	96.96%
Being an HIV-infected patient with a CD4+ cell count of 200–350/mm^3^, regardless of vaccination status or history of SARS-CoV-2 infection	MODERATE	MODERATE	100%	78.78%
Being a patient with HIV infection and poorly maintained virological control of HIV, regardless of vaccination status or history of SARS-CoV-2 infection	VERY LOW	WEAK	100%	84.84%
Being an HIV-infected patient with CD4+ cell count > 350/mm^3^ is NOT considered to be an increased risk factor	LOW	WEAK	100%	100%
Being a patient with HIV infection and having had a CD4+ nadir <200/mm^3^ is NOT considered to be an increased risk factor	HIGH	STRONG	100%	78.78%

* A cut-off point for reaching consensus was established when 80% of the experts ticked maximum agreement (on a scale of 1 to 9: score ≥ 7).

**Table 11 viruses-15-01449-t011:** Therapeutic recommendations for patients with COVID-19 and primary or other secondary immunodeficiencies.

Profile of Patients with an Increased Risk Factor for Severe Forms of COVID-19	Quality of the Evidence	Strength of the Recommendation (Therapeutic Action to Prevent Progression to More Severe Stages of the Disease)	Meeting (n = 7) *	Online Consensus (n = 33) *
Being a patient with primary congenital immunodeficiency, regardless of vaccination status or history of SARS-CoV-2 infection	LOW	WEAK	100%	100%
Being a patient with functional or anatomical asplenia, regardless of vaccination status or history of SARS-CoV-2 infection	LOW	WEAK	100%	93.93%

* A cut-off point for reaching consensus was established when 80% of the experts ticked maximum agreement (on a scale of 1 to 9: score ≥ 7).

**Table 12 viruses-15-01449-t012:** Therapeutic recommendations for patients with COVID-19 and autoimmune diseases.

Profile of Patients with an Increased Risk Factor for Severe Forms of COVID-19	Quality of the Evidence	Strength of the Recommendation (Therapeutic Action to Prevent Progression to More Severe Disease)	Meeting (n = 7) *	Online Consensus (n = 33) *
Being a patient with certain systemic or immune-mediated autoimmune diseases **, depending on their stage and treatment modality, regardless of vaccination status or history of SARS-CoV-2 infection	LOW	MODERATE	86%	81.81%
Being a patient on chronic glucocorticoid therapy, regardless of vaccination status or history of SARS-CoV-2 infection	MODERATE	MODERATE	86%	90.90%
Being a patient on anti-CD20 treatment, regardless of vaccination status or history of SARS-CoV-2 infection	HIGH	STRONG	100%	100%
There is NOT enough published evidence to consider patients on treatment with other immunosuppressants than those previously indicated (e.g., JAK or TNF inhibitors) as an increased risk factor.	LOW	STRONG (as there is no evidence to recommend its use)	100%	87.87%

* The cut-off point for reaching consensus was established when 80% of the experts ticked maximum agreement (on a scale of 1 to 9: score ≥7). ** Apart from the diagnosis of an IMID, it may also be influenced by the type of immune-mediated inflammatory disease per se, its stage or degree of involvement and extent, particularly additional comorbidities, and above all, the type of disease-modifying treatment, biological or non-biological, and the family to which the latter belong, as well as their target and mechanism of action.

## Data Availability

All published materials reviewed and all documents generated during the elaboration of this expert consensus are available upon request.

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
