# Peer review of "Expert Consensus: Main Risk Factors for Poor Prognosis in COVID-19 and the Implications for Targeted Measures against SARS-CoV-2"

_viruses, 2023, doi:10.3390/v15071449_

Round 1

Reviewer 1 Report

I think this paper is a valuable contribution to the field of covid-19 treatment, as the authors did a good job at combining all the evidence regarding risk factors for progression to severe illness and they made clear recommendations. This is very commendable and it was carried out well, from what I could deduct from the paper. 

However, I do not feel this is truly a systematic review. Although some steps along the process were indeed systematic (search strategy, two independent assessors), it is telling that the majority of the included papers were identified through more informal (scoping, snowball) review. I think this topic does not lend itself easily to a systematic review; the number of included papers was simply too large to assess the papers in a systematic way. 

For example, although the authors use the PRISMA guidelines, they incorrectly state that they included the details of each included paper. They did not; in fact, they did not even cite all the included papers in the reference list (278 included papers according to figure 2, but only 251 papers are cited, and this includes the (many) cited papers from the introduction!). Of course it is impossible and impractical to cite details for all 278 papers; but then maybe the scope was too large for a systematic review? Quality of the papers is also not assessed, again, quite understandably given the body of research, however therefore it just doesn't quality as a systematic review.

Then what is it? I would personally recommend the authors to change the name and scope of the paper, and instead of calling it a systematic review, I would describe this as an "expert consensus". The document reads as an expert consensus guideline, and this is probably what it is. 

Having said that, I would also make the following recommendations:

- how were these experts identified? Was this an international panel?
- not all of these experts were co-authors (37 experts; only 20 authors); where can I find a list of the experts? I also note that all of the authors are Spanish; was this a Spanish expert panel? How was the expertise determined?
- Figure 2 is not entirely correct; I would recommend to change the following: "Records sought for retrieval" --> "Records FOUND for retrieval and assessed for eligibility (n = 239)"; the arrow to "reports not retrieved (n = 32) should be higher, next to the arrow or box above; and so on...
The rest of the flow diagram is not quite right either; please have a look to see whether this can be improved to be more logical (the numbers are correct but it looks incorrect as the order and terminology used are unclear)
- The introduction is much too long (4.5 pages!); this should be more concise. The authors have already done a mini-review of the evidence in the introduction, and then everything comes back in the main text. 

Suggested corrections:

- p7 has an error in the references (line 325)
- Annex 1 --> I would call this Table 1, it's not really an annex, it's actually the core of your review

Abstract: line 38 "has slowly may the pandemic to wane" --> "has slowly made the pandemic to wane"? 

line 40-41: I think the internationally accepted de-abbreviation of covid-19 is coronavirus disease 2019, and not the (overly long?) coronavirus infectious disease detected in 2019. 

line 42: I would change "provided" to "administered"

Author Response

Dear reviewer,

we appreciate the improvements you have introduced in the manuscript. Firstly, the quality of the English language has been edited as suggested (marked in blue). Also all your additional comments have been addressed as follows (marked in red):

Describe the review as an "expert consensus": The type of review has been changed accordingly. The title and methodology has been modified according to the character of the review.

How were these experts identified? Was this an international panel?: All panelists were experts working in Spanish centers that were selected by the coordinators of the project (FJC, PB and MS) considering the area of specialization and publications, communications and participation in scientific meetings about COVID-19. A total of 20 panelist with a greater level of participation author the manuscript; another 17 experts are part of the reviewers and have been named in the addenda of investigators.

Figure 2 is not entirely correct: the figure has been corrected.

The introduction is much too long: the Introduction has been shortened.

Error in page 7, line 325: The mistake has been corrected.

Annex 1: It has been renamed as Table 1.

Our sincere thanks,

Reviewer 2 Report

Great and important research to generate expert consensus- 280 based recommendations to establish the patient profiles that can benefit most from the 281 use of AVTM against SARS-CoV-2

The research is well written and I appreciate the efforts made by the authors, but some modifications need to be made.

 the results section should be renamed into results and discussion and the conclusion section should be added.

Author Response

Dear reviewer, the manuscript has been modified according to your recommendations. The quality of the English language has been improved. The Results section has been renamed as Results and Discussion. A Conclusion has been added. Many thanks,
